# Development of an Effective Information Media Using Two Android Robots

**Toshiaki Nishio [1,]*  , Yuichiro Yoshikawa [1], Kohei Ogawa [1] and Hiroshi Ishiguro [1,2]**

1   Graduate school of Engineering Science, Osaka University, Osaka 560-8531, Japan
2   Intelligent Robotics and Communication Laboratories, ATR, Kyoto 619-0237, Japan
*   Correspondence: nishio.toshiaki@irl.sys.es.osaka-u.ac.jp

**Abstract:** Conversational robots have been used to convey information to people in the real world. Android robots, which have a human-like appearance, are expected to be able to convey not only objective information but also subjective information, such as a robot's feelings. Meanwhile, as an approach to realize attractive conversation, multi-party conversation by multiple robots was the focus of this study. By collaborating among several robots, the robots provide information while maintaining the naturalness of conversation. However, the effectiveness of interaction with people has not been surveyed using this method. In this paper, to develop more efficient media to convey information, we propose a scenario-based, semi-passive conversation system using two androids. To verify its effectiveness, we conducted a subjective experiment comparing it to a system that does not include any interaction with people, and we investigated how much information the proposed system successfully conveys by using a recall test and a questionnaire about the conversation and androids. The experimental results showed that participants who engaged with the proposed system recalled more content from the conversation and felt more empathic concern for androids.

**Keywords:** human robot interaction; android robot; passive social conversation; multiple conversation robots

---

## 1. Introduction

In recent years, communication robots have become popular in the real world [1]. To promote their use in the current society, the capability of communication in robots should be improved so that users can interact with robots more easily. One approach to realizing easy communication with a user is using humanoid robots. For example, robots have been used as communication media that provide information autonomously to visitors in public spaces such as a science museum [2] and a shopping mall [3]. Humanoid robots that resemble the appearance of human beings are called android robots. Owing to their appearance, an observer who sees android robots considers android robots as human for a few seconds [4]. Android robots can present a stronger sense of existence than other media such as video, or a speakerphone [5]. Android robots can effectively represent human-like mental states, including subtle emotion, by using nonverbal communication [6]. Therefore, it is argued that android robots can become influential conversation media that can effectively convey information [7].

In previous work on the conversation of agents, voice has been widely adopted as one of the most accessible and most universal modalities in many applications, such as an application for weather forecasts [8], for the guiding of buses [9], and for providing information to visitors in a science museum [10]. However, voice recognition in the real world has still been a formidable challenge [11]. Furthermore, especially in robot applications, it becomes more difficult because a human would often speak in less formally correct ways, such as in dialects, toward the human-like agent. Adopting a very human-like appearance in the android robot also causes a risk called the adaptation gap [12]: humans

tend to be easily disappointed by the robot's utterance because they have heightened expectations for a human-like, contextually natural response due to the appearance of human-like organs corresponding to voice production, i.e., mouth and throat.

To adopt verbal communication in robot applications without suffering from potential failure and disappointments, a specific form of communication called multi-party conversation has been developed in the field of human–robot interaction [13]. There is a study of robots as a passive social medium, where robots provide information for people with dialogue between two robots while not interacting with people directly [14]. It was reported that passive social robots succeeded in getting more attention from observers than a single robot receives alone. People who observe a scene in which two robots communicate with each other are more likely to treat those robots as if they are human and evaluate their dialogue as more understandable than they would for robots without communication between them [15]. However, without listening to a human response, it is difficult for robots to keep a human paying attention to the conversation for an extended period. Arimoto et al. showed that inter-robot turn-taking triggered by a human response contributes to maintaining the sense of conversation, even though the robots produced pre-scripted utterances and ignored the content of the human response [13]. However, the extent to which their conversation was successfully conveyed to humans has not been examined, especially in the case of affective content. Therefore, we constructed and examined a scenario-based, semi-passive conversation system using two androids that basically talked to each other and sometimes actively listened for human answers to their questions, including affective ones. To validate the usefulness of semi-passive social androids as a medium to provide information for people, a comparative experiment with passive social androids was conducted. Specifically, we evaluated the feasibility of semi-passive social androids to transmit the feelings of androids (subjective information) and the contents of the conversation (objective information) to people. To confirm the functionality of communicating subjective information, we prepared a script that had some statements for making participants feel empathic concern with androids and asked for their impressions of the androids through a questionnaire. Additionally, to confirm the functionality of communicating objective information, we added general information about androids in the script and measured how much information the participants could remember with a recall test. The between-subjects design experiment was conducted, where the subjects talked with either semi-passive social androids or passive social androids for ten minutes. After the talk, participants evaluated the androids and the conversations with a recall test and a questionnaire. From the result, it was verified that the semi-passive social condition was assessed higher than the passive social condition based on the information memorized by the subjects and the emotions conveyed and evoked by the androids.

## 2. Materials and Methods

This section describes details of the semi-passive dialogue system using two androids, as well as the experiment to validate its effect on the conveyance of the objective and subjective information.

### 2.1. System Overview

The dialogue system developed in this research was a scenario-based verbal communication system. The system supposed multi-party conversation among two androids and a person. Two androids normally led the conversation by talking with each other and sometimes engaged the person in the conversation by asking his/her opinion or experience. After receiving an answer from the person, the androids recognized it and responded to it. Androids spoke and moved according to directions written in a prepared script, which included the contents of speech and behavior for each android. Some templates for responding to the person were also written in the script, and the system selected a suitable template as a response.

Figure 1 shows an overview of the system. The system was comprised of four modules: the speech recognition module, the text-to-speech module, the androids' behavior control module, and the

scenario manager module. Androids could vocally communicate with people by integrating these modules together. The function of each module was as follows:

- Speech recognition module: This module's function was to capture the voice data of a person and convert the voice data to text.
- Text-to-speech module: This module's function was to convert the text received from the scenario manager module into voice data and playing the resulting voice data.
- Androids' behavior control module: This module controlled the body movement of the androids, such as the mouth movement for utterance, head or eye motion for looking at people or the other robot, breathing motion, and emotional expression.
- Scenario manager module: This module was used for reading the script and controlling other modules by sending order sequences.

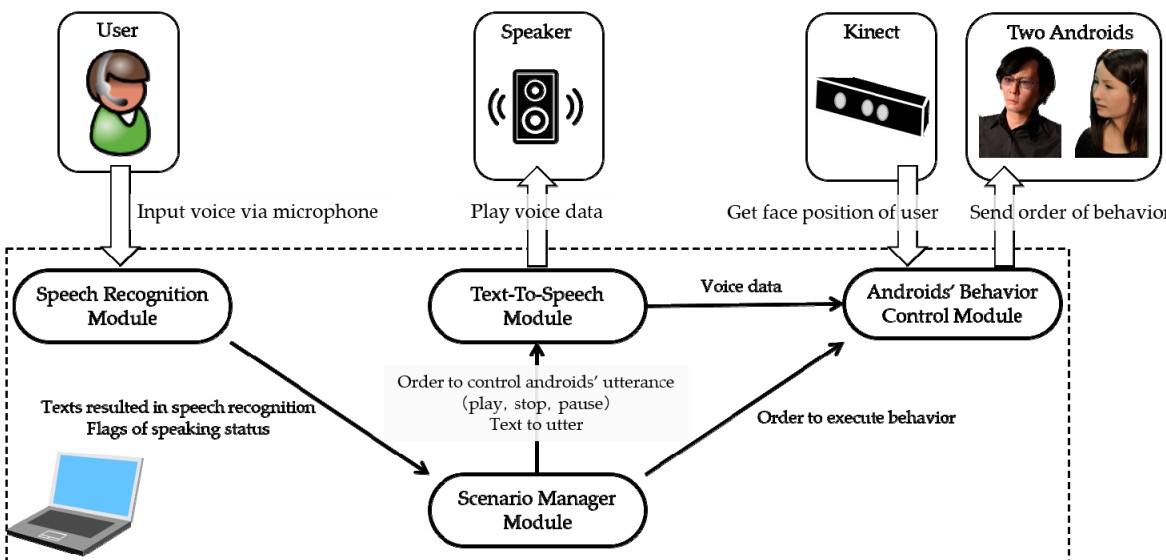

**Figure 1.** System diagram.

### 2.1.1. Android: The Robot Having Human-Like Appearance

An android robot is a humanoid robot that has a very human-like appearance. For the conversation system, we employed a female android called Geminoid F (Figure 2a) and a male android called Geminoid HI4 (Figure 2b). Both androids were developed in a collaborative project between Osaka University and Advanced Telecommunications Research Institute International (ATR). These androids have skin made of silicon rubber, and the shape of their skin was molded from an actual human. Because the tactile sensation and texture of the surfaces are very close to that of a human, they can convey a human-like presence. In addition, these androids have many degrees of freedom (DOFs) in their faces (Geminoid F: 7 DOFs, Geminoid HI4: 8 DOFs), with enough DOFs on their face allowing them to convey a natural human-like presence. Since the androids are driven by air compression actuators, they can move softly and show their behavior without producing loud noises. Due to these factors, we concluded that Geminoid F and Gemenoid HI4 were suitable for showing human-like communication with people. On the other hand, neither F nor HI4 have any sensor to get environmental information inside their body, and therefore, additional external sensors were needed to obtain the voices and positions of people. Each robot had a speaker inside its body; however, a sound played from the speaker was not clear. Thus, we designed external speakers for each android.

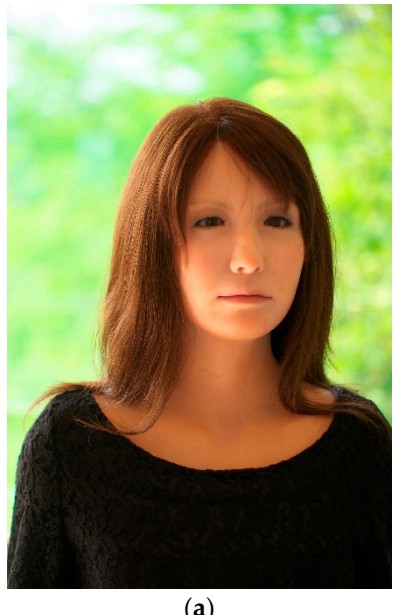
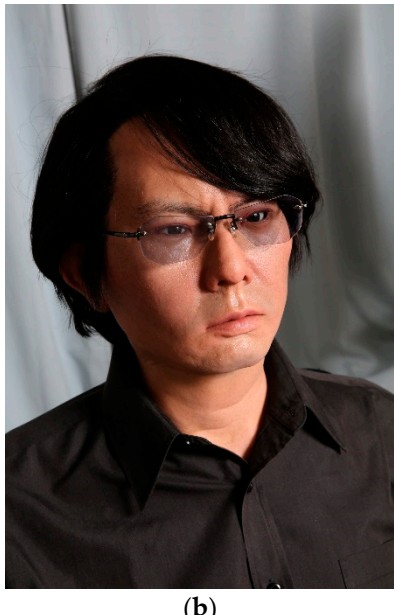

(**a**)  (**b**)

**Figure 2.** Appearance of Geminoids: (**a**) Geminoid F: female android; (**b**) Geminoid HI4: male android.

### 2.1.2. Speech Recognition Module

The speech recognition module converts voice data from people into text and sends the recognized text to the scenario manager module. A non-directional close-talking microphone is used to capture voices, and Dragon Speech 11 (Nuance Communication Inc., Burlington, MA, United States) is used for speech recognition. The module also detects utterance sections using a threshold of sound pressure. Specifically, the module measures an average of sound pressure sampling of 44.1 kHz for every millisecond. When an average pressure over the specified threshold has been detected for more than 500 milliseconds continuously, the module determines that people are in talking status. When an average pressure under the threshold has been detected for more than 1500 milliseconds in the talking status, the module determines that the person has finished speaking. The module also sends flags to represent the talking and talk-end status to the scenario management module. To avoid detecting noises, the module sends text recognized by Dragon Speech only during the talking status.

### 2.1.3. Text-To-Speech Module

The text-to-speech module converts text received from the scenario manager module to sound data and plays it. Using the AITalk Custom Voice (AI Inc., Tokyo, Japan) as a voice synthesizer, the module can do real-time synthesis, in which the synthesizing of voice data and playback on a device are executed in parallel. The module receives three types of orders with the text from the scenario manager module: start playing, stop playing, and pause playing. The module can not only stop/pause playing voice data immediately but also optionally stop/pause it at the end of a current phrase. The module has variable parameters for the start playing order, including volume, speed, pitch, range, length of short pauses in the sentence, length of long pauses in the sentence, and the length of the end pause of the sentence.

### 2.1.4. Androids' Behavior Control Module

The Androids' Behavior Control Module controls the actuators of androids to express various behavior such as emotional expression, mouth movement with synchronized voice, and motion to make an android look at people or the other android. The behavior of the android is described in a motion file, which contains the positions of each joint at 50 milliseconds intervals called frames. The position of a joint is set using a value from 0 to 255. Androids execute idling behaviors, which are

minimal behaviors like blinking, as well as breathing, which is expressed by slight movements of the shoulders, waist, and neck. Moreover, an android can express these behaviors at any time by executing prepared motion files. Some motions for which preparation is difficult, like looking at the person's face, are realized using sensors. When looking at a face, the face position is detected in three-dimensional space using Kinect for Windows v2 by Microsoft, and is associated with the three axes of the neck of the android. Values of these three axes are transformed from the face position using a projection matrix calculated via calibration using the least squares method with 16 pairs of position of the axes of the neck and the position of the detected face. Regarding lip motion synchronized with voice, the motion is generated using formant information included in the voice [16].

### 2.1.5. Scenario Manager Module

The scenario manager module controls the conversation and behavior of two androids by following a prepared script. In the script, the details of a dialogue, such as speech contents, the behavior of the two androids, and the timings of speech recognition, are specified, and the module sends orders to the other modules sequentially. Moreover, the script also has templates to be used as a response of an android to utterances from people at every scene, for which an android asks people something and the person may utter. Androids encourage the person to utter by asking him/her a yes-or-no question. When the person utters something, the module identifies the utterance as positive or negative, and the androids generate a response using a template for each category. If the person has not spoken anything or the speech recognition module fails to recognize speech, the module selects a template for an ambiguous response.

The module has a classifier to categorize the utterance of the person and learns it from training data, which is constructed using pairs of a recognized text of the utterance of people and a label of it. After morphological analysis of training data, vectors of bag of words (BOW) are created. Incidentally, stop words included in SlothLib [17], which is a programming library, are removed from the data. Also, high-frequency words included in the top 0.9% of the data and low-frequency words appearing only two times are removed. Additionally, to reduce the amount of sampling used for categorization, the dimension of BOW vectors is diminished until reaching the number of classes by using Latent Semantic Indexing (LSI). The module learns support vector machine (SVM) using BOW vectors and labels.

Morphological analysis is performed using JUMAN [18] which is a Japanese morpheme analysis program. For SVM, Scikit-Learn [19] is used, which is an open source machine learning library. Gemsim [20] is used for creating vectors of bag of words, and is a topic model library in Python.

After selecting a template, the module creates a text response by inputting words included in the person's utterance. What is extracted from the person's utterance are words with polarity. To extract polarity words, Japanese Sentiment Polarity Dictionaries (Volume of Verbs and Adjectives) ver. 1.0, created by Inui-Okazaki laboratory at Tohoku University, is used.

### 2.2. Experiment

This section describes a subject experiment between participants conducted to reveal the effectiveness of semi-passive social androids as communication media to convey objective information and subjective information. The semi-passive social androids tried to engage the participants by repeatedly giving directions to them in the conversation. This was expected to help the participants to remain engaged in and concentrating on the conversation. This engagement and concentration were expected to help the participants to recall the message in the conversation. Meanwhile, it was clear the messages when the messages were being directed to the participants because the android uttered toward the participant, and the participants were expected to feel the stronger will of the androids conveying them, and as a result, the messages were expected to be recognized as stronger. Accordingly, with respect to these expected effects of the semi-passive social androids, we examined three hypotheses in this experiment: (i) Participants will recall more objective information given in the semi-passive social conversation of two androids than in the passive social one. (ii) Participants

will feel the subjective messages as being stronger in the semi-passive social conversation than in the passive social conversation. (iii) Participants will be moved to follow the subjective messages in the semi-passive social conversation more than in the passive social conversation. In addition to the three hypotheses, two other points, namely the degree of participant engagement with the conversation and the difficulty of the conversation, were surveyed to confirm that the system was able to run as intended and the experiment was conducted as expected.

### 2.2.1. Participants

The current study was approved by The ethics committee for research involving human subjects at the Graduate School of Engineering Science, Osaka University. The participants were 28 university students (14 male and 14 female students), whose ages ranged from 19 to 22 years old. They were randomly assigned to two conditions. In the passive condition, seven males and seven females participated in conversation with two androids that talked and directed their gaze to each other but did not do so to the participant. In the semi-passive one, the remaining seven males and seven females participated in the conversation with two androids that talked and directed their gaze not only to each other but also to the participant. No participants had any experience directly interacting with the androids before the experiment.

### 2.2.2. Apparatus

In the experimental room, the participants faced two androids across a table. The distances between the two androids and the participant formed an equilateral triangle with 1.2-m sides (Figure 3). Figure 4 shows a scene of the experiment.

The topic of the conversation was related to android robots and the Intelligent Robotics Laboratory where they were developed. We designed an approximately ten-minute scenario for the conversation to introduce both the subjective and objective information. To verify how efficiently the subjective information was conveyed, we included utterances by which the androids explained their situation and how they felt. For example, "Visitors touch my body as they like. I would like them to refrain from it a little more." and "I am seldom quickly repaired (by the laboratory members) when I am broken. Don't you think it's terrible treatment?" Note that, in the semi-passive social condition, when they mentioned "you," they looked at the participant. In contrast, in the passive one, they did not look at him or her. Meanwhile, to verify how efficiently the objective information was conveyed, we included utterances with numbers or proper nouns to introduce facts about the laboratory and the androids. For example, "In fact, our bodies are driven by air pressure at about 0.7 MPa." Figure 5 shows a part of the conversation used for the experiment. In addition to the script of the passive social condition, nine utterances toward the participant were included in the conversation used for the semi-passive condition. Note that, to avoid providing further information by these nine utterances, the androids were limited to utterances asking about the participant's agreement or opinions in relation to the androids' conversation.

The responses from the participants were classified into the four categories: no answer, positive answer, negative answer, and other. This allowed the androids to choose an appropriate but brief response without providing any further information. To categorize the participant's answer as one of the four types, the system was trained before the experiment. For this, we constructed another system that could select a response for the androids using the Wizard of Oz method and collected morphemes as training data via conversations between eight males and the system.

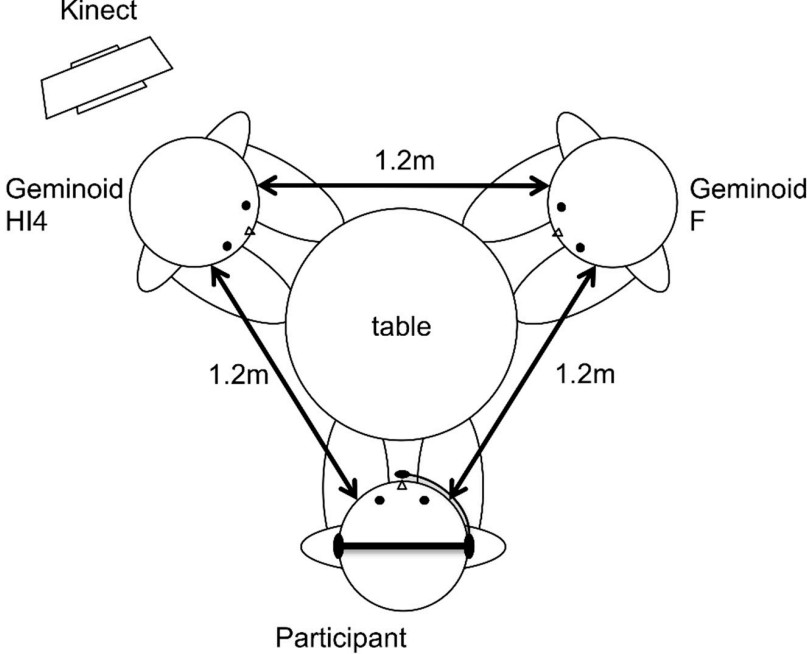

**Figure 3.** The environment of the experiment.

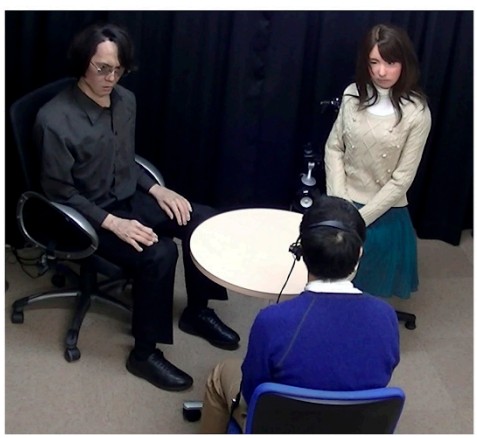

**Figure 4.** A scene of the experiment.

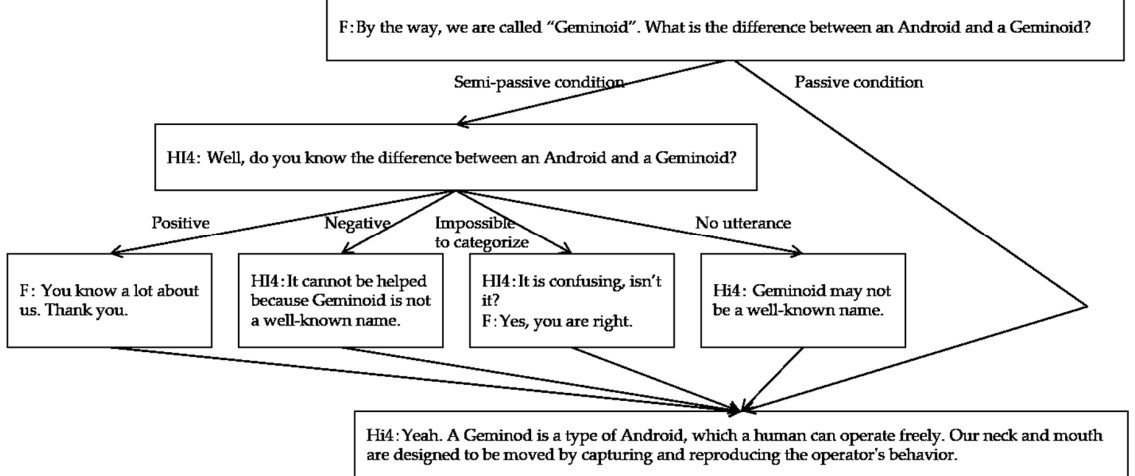

**Figure 5.** Example of the conversation.

### 2.2.3. Procedure

The experiment was conducted using the following procedure, one participant at a time. First, a participant waited in an anteroom until the start of the experiment. At a specified time, an experimenter explained an overview of the experiment in the anteroom and got the participant's consent regarding participation. After that, the experimenter provided instruction regarding the procedure of the experiment. In the instruction, the participant was requested to listen to the conversation between the androids. However, if the androids asked them questions during the conversation, the participant was permitted to answer it. Next, the participant put on a headset and configured the microphone using a function of Dragon Speech, which is an automatic speech recognition software. The software can adjust the volume level and check the quality of voice input by having the participant read a few prepared sentences aloud. Then, the participant moved to an experimental room containing the two androids. The participant sat across the table from the two androids. When the participant took their seat, the two androids started to talk each other. The conversation lasted for around 10 min, and when the conversation finished, the experimenter called out to the participant, directing them to go back to the anteroom. The participant removed the headset and answered a questionnaire and a recall test about the conversation. Lastly, the experimenter interviewed the participant.

### 2.2.4. Measurements

First, we checked whether the participants were adequately stimulated from the following two points of view. Namely, we checked whether the contents of the conversation presented in the experiment were too difficult for the participants to understand and whether the participants in the semi-passive condition were able to engage in the conversation. Regarding the difficulty of the conversation, it was checked whether the contents of conversation made sense (D-1) to the participants. Meanwhile, it was supposed to not be too difficult for participants (D-2) to avoid a floor effect in evaluating how much objective information was conveyed to participants.

D-1. Were you able to understand the conversation entirely?
D-2. Were you able to consent the conversation entirely?

Regarding the degree of engagement to the conversation, it is important for the system to correctly handle the replies from the participants so as to prevent them from feeling ignored to provide the experience of semi-passive conversation. If the system failed to do that, it was considered that the androids' attitude toward engagement in the conversation with the subjects, i.e., the feeling of being given attentive (E-1) and interested (E-4), as well as the participants' attitude to engage in the conversation with the androids, i.e., the motivation to speak (E-3) and to join the conversation (E-2), would not be recognized. Note that only the participants in the semi-passive condition were evaluated for this as a comparison to the passive condition in this regard was considered to be unfair because the androids never paid their attention to the participants in the passive one. The degree of engagement with the conversation was surveyed using the following questions:

E-1. Did you feel that the female android and the male android paid attention to you?
E-2. Did you feel that you were able to join the conversation?
E-3. Did you feel that you could give your opinion to the female android and the male android?
E-4. Did you feel that the female android and the male android were interested in your opinion?

To verify the first hypothesis regarding objective information, we conducted a recall test to confirm how much the participants memorized the content of the conversation. We created ten questions to ask about the contents that the androids conveyed in the conversation. For example, the participants were asked: "How many androids existed in this laboratory?", "To which country did the female android want to go?", and so on. We counted how many questions the participants could correctly answer. To easily judge whether the answers were correct, questions were chosen so that the correct answers could be described as a single word.

The second and third hypotheses are related to the effective conveyance of the subjective information. The second one concerns whether participants regarded messages from the androids more strongly in the semi-passive condition than in the passive one. To verify this, the strength of messages from the androids about the subject's experience in the conversation was evaluated by using the below questionnaire:

2-1.　Androids requested you to help them leave the bad situation where they got a rough deal in the laboratory. Did you think of telling the laboratory members the request from the android?

2-2.　Did you feel that the female android and the male android had firm beliefs?

2-3.　Did you feel that you were urged strongly by the female android and the male android?

2-4.　Did you feel that the statements of the female android and the male android were persuasive?

Furthermore, as predicted in hypothesis (iii), the participants were expected to be moved to follow the subjective messages in the semi-passive social conversation than in the passive social conversation. To verify this, we measured how much empathic concern the androids elicited from the subjects by using a questionnaire. In this experiment, we focused on empathic concern as a typical example of emotional response because the androids talked about their story to effect empathic concern in the conversation. Specifically, the following two questions were used:

3-1.　Androids said in the conversation that they were often told that they were scary or spooky. Did you feel that it was cruel?

3-2.　Androids said in the conversation that they were carried as baggage. Did you feel that it was cruel?

A seven-point Likert scale was adopted for questions about the strength of the message, empathic concern, the difficulty of contents, and engagement, which ranged from 1: Strongly disagree, 2: Disagree, 3: More or less disagree, 4: Undecided, 5: More or less agree, 6: Agree, to 7: Strongly agree.

## 3. Results

### 3.1. Manipulation Check

Regarding the difficulty of the conversation, no participant in either condition marked under 4 for the two questions (QD-1 and QD-2), which was interpreted as indicating that the contents of the conversation were appropriately prepared so as not to be too difficult. Regarding the degree of engagement to the conversation, the median scores obtained from the participants attending the semi-passive conversation for four questions (QE-1, QE-2, QE-3, and QE-4) were not less than intermediate points, which was interpreted as indicating that the system could provide the experience of conversation with a moderate level of engagement.

### 3.2. Recall Test for Hypothesis (I)

We evaluated the difference between the semi-passive condition and the passive condition using a recall test and the questionnaire. Figure 6a shows the average number of questions that the participants correctly answered. Mann–Whitney's U test was used to compare the scores of the two conditions. The result shows that the median of the number of correct answers of the semi-passive condition (Mdn = 8) is significantly higher than that of the passive condition (Mdn = 7) ($U = 54.5$, $p < 0.05$).

### 3.3. Evaluation of Questionnaire for Hypotheses (II) and (III)

Figure 6b,c show the questionnaire results. Medians of each question between the two conditions were compared using Mann–Whitney's U test. Figure 6b shows scores of four questions about the strength of subjective message (Q2-1, Q2-2, Q2-3, and Q2-4), and no significant difference was suggested. Figure 6c shows scores from the two questions about empathic concern from participants toward the androids (Q3-1 and Q3-2). The result of Q3-1 shows that the score of the semi-passive condition (Mdn = 5.5) was higher than that of the passive condition (Mdn = 4.5) with a marginal difference

(U = 57.5, $p < 0.1$). The result of Q3-2 shows that the score of the semi-passive condition (Mdn = 4.5) was significantly higher than that of the passive condition (Mdn = 3) (U = 48.5, $p < 0.05$).

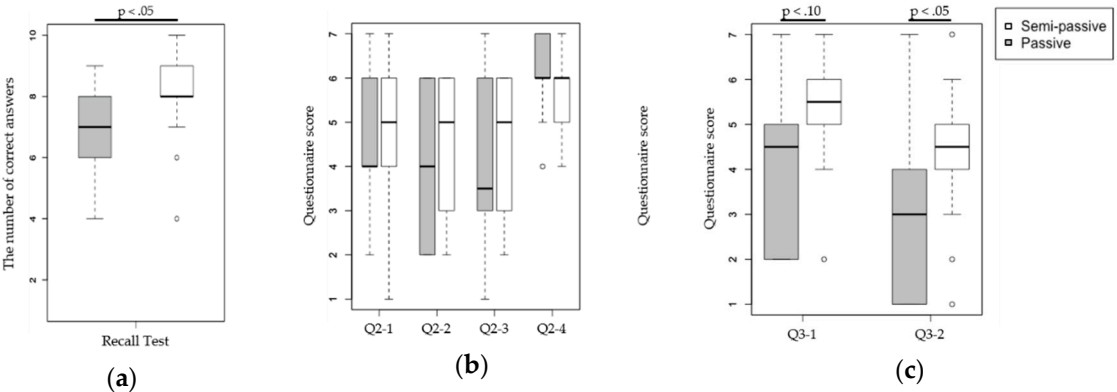

**Figure 6.** The result of (**a**) the recall test, and the (**b**) strength of subjective message, and (**c**) empathic concern questionnaires.

## 4. Discussion

During the analysis of the recall test, we found that the participants remembered more words of the conversation in the semi-passive condition than in the passive one, which supports the first hypothesis. In the experiment, the participants attended a 10-min conversation where the androids asked them something roughly once every minute in the semi-passive condition. The questions in the recall test were chosen so that the answer words appeared between these interaction blocks of asking and answering in the semi-passive condition. In other words, the questions were independent of what the androids asked. Therefore, the interaction block was considered to encourage participants to remember not only the words that appeared in the block, but also in the entire conversation.

Regarding participants' evaluation of the strength of message, we did not find any improvement in the semi-passive condition compared to the passive one, which does not support the second hypothesis. In other words, the semi-passive form of the conversation did not effectively contribute to how much the participants were convinced by the subjective message from the android. On the other hand, higher empathic concern toward the androids was observed in the semi-passive condition than in the passive one, supporting the third hypothesis. In other words, the participants were moved to follow the misery situation of the androids. These two potentially inconsistent results may imply that the subjective message from the androids in the semi-passive conversation was successfully conveyed not through being felt that they were strongly requested but rather through spontaneously evoking the subject's empathic concern.

It is worth considering how the current result is limited by the fact that we tested only with one scenario including a subjective message from the androids. The remaining question is whether the improvement in recall performance is also maintained in a scenario without such a message, which should be beneficial for the general purpose of information conveyance. Therefore, a further experiment should use content with only objective information. On the other hand, even if it is limited to cases with subjective aspects, the results of this study should still be good news for information media because the development of affective applications have been considered [21]. However, the current result might be limited to the android robot, which is very human-like. Therefore, it is also worth investigating how much the information media should be human-like so that the effect of the semi-passive form of conversation can be utilized.

## 5. Conclusions

In this paper, we proposed efficient media to convey information based on what we call semi-passive conversation. To test the effect, a conversation system using two androids was

implemented, which consisted of a scenario-based conversation system having a function of sometimes briefly listening to the user's agreement or disagreement to the conversation. Regarding objective information, the result of the recall test showed that the participants who attended the semi-passive conversation remembered more words in the conversation than those who attended the passive conversation which involved the same content except for the interaction blocks that checked the participants' agreement or disagreement. Regarding subjective information, the participants showed stronger agreement for the action of the androids, which was interpreted as indicating that spontaneously empathic concern was evoked more in the semi-passive conversation than in the passive one. Further experiments changing the subjective quality of the contents and human-likeness of the robots are important for future work.

**Author Contributions:** Conceptualization, T.N., Y.Y., K.O. and H.I.; methodology, T.N., Y.Y. and K.O.; software, T.N.; validation, T.N.; formal analysis, T.N.; investigation, T.N.; resources, Y.Y., K.O. and H.I.; data curation, T.N.; writing—original draft preparation, T.N.; writing—review and editing, Y.Y. and K.O.; visualization, T.N.; supervision, Y.Y. and K.O.; project administration, H.I.; funding acquisition, H.I.

**Funding:** This work was supported by the Japan Science and Technology Agency (JST) Exploratory Research for Advanced Technology: ERATO, grant number JPMJER1401, Japan.

**Conflicts of Interest:** The authors declare no conflict of interest.

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
