# Peer review of "Development of an Effective Information Media Using Two Android Robots"

_applsci, doi:10.3390/app9173442_

Round 1
Reviewer 1 Report
This article shows that information is conveyed more successfully in the semi-passive setting than in the passive setting of android conversation system.
This reviewer suggests that the authors differentiate more clearly between hypothesis ii (Participants recognize the conveyance of subjective information more strongly, lines 200-201) and iii (Participants drew a greater emotional response congruent with the subjective information, line 202). There are two related problems. Firstly, the authors test hypothesis ii by asking mostly how the participants “feel” (questions 2-2 to 2-4, lines 281-283), and assume positive answers mean “it moves the participants (line 284).” A discrepancy between the hypothesis (recognition of the conveyance of subjective information) and its testing (feelings of the participants), therefore exists. Secondly, the actual testing of hypothesis ii (feelings of the participants towards attitudes and beliefs of the androids) cannot be fully differentiated from hypothesis iii (empathic concern for the androids).
From the problems stated above, the inconsistent results of hypotheses ii and iii (lines 360-363) seems to be more interesting because the actual testing of hypothesis ii and iii might be testing similar or even the same thing. If it is possible, showing some parts of the conversation between the androids and the participants, especially those related to hypotheses ii and iii, may be helpful not only for readers to have a clearer understanding of how hypotheses ii and iii and their testing are about, but also for the authors to discuss the inconsistency of the results.
Author Response
Dear Reviewer
Thank you for your careful review.
Please see the attachment for our responses to your comments.
We believe that the manuscript was improved by the your suggestions.
Best regards

Reviewer 2 Report
This paper investigated a problem that introduces semi-passive conversation into human-robot interaction that involves two android robots and one human. An experiment has been conducted to support that the proposed semi-passive conversation system is effective and more efficient in conveying information and empathic engagement.
The validity of the experiment is based on the performance of the conversational system. It is described as a prepared script within different scenarios. It would be nice to provide more details, like the size of scripts, and how scenarios are selected, with a few more examples.
The section 2.2.4 measurements and section 3 result can be rewritten for a clearer structure, in which hypotheses are followed by analyses. Also, the current hypotheses are very weak, for example, whether android paid attention to participants. Something is missed in connecting them with the claims in the efficiency of information conveying and empathic engagement. It is very hard to find out what the authors want to highlight in the current writing.
Author Response

(The authors gave the same response as above.)
